# MKK4 Inhibitors—Recent Development Status and Therapeutic Potential

**DOI:** 10.3390/ijms24087495

**Published:** 2023-04-19

**Authors:** Leon Katzengruber, Pascal Sander, Stefan Laufer

**Affiliations:** 1Department of Pharmaceutical/Medicinal Chemistry, Institute of Pharmaceutical Sciences, Faculty of Sciences, University of Tuebingen, 72076 Tübingen, Germany; leon.katzengruber@uni-tuebingen.de (L.K.); pascal.sander@uni-tuebingen.de (P.S.); 2Cluster of Excellence iFIT (EXC 2180) ‘Image-Guided & Functionally Instructed Tumor Therapies’, Eberhard Karls Universität Tübingen, 72076 Tübingen, Germany; 3Tübingen Center for Academic Drug Discovery, Auf der Morgenstelle 8, 72076 Tübingen, Germany

**Keywords:** MKK4, MEK4, MAP2K4, MAPK, kinase inhibitors, drug design, cancer, liver regeneration, liver failure, clinical trials

## Abstract

MKK4 (mitogen-activated protein kinase kinase 4; also referred to as MEK4) is a dual-specificity protein kinase that phosphorylates and regulates both JNK (c-Jun N-terminal kinase) and p38 MAPK (p38 mitogen-activated protein kinase) signaling pathways and therefore has a great impact on cell proliferation, differentiation and apoptosis. Overexpression of MKK4 has been associated with aggressive cancer types, including metastatic prostate and ovarian cancer and triple-negative breast cancer. In addition, MKK4 has been identified as a key regulator in liver regeneration. Therefore, MKK4 is a promising target both for cancer therapeutics and for the treatment of liver-associated diseases, offering an alternative to liver transplantation. The recent reports on new inhibitors, as well as the formation of a startup company investigating an inhibitor in clinical trials, show the importance and interest of MKK4 in drug discovery. In this review, we highlight the significance of MKK4 in cancer development and other diseases, as well as its unique role in liver regeneration. Furthermore, we present the most recent progress in MKK4 drug discovery and future challenges in the development of MKK4-targeting drugs.

## 1. Introduction

Kinases have been one of the main challenges of drug discovery in the past 20 years as they regulate nearly all aspects of cell life and alterations that cause cancer or other diseases in a wide range [1,2]. To date, more than 70 kinase inhibitors have been approved, of which a majority are effective against various cancers and only a few against other diseases such as rheumatoid arthritis [2]. However, less than half of the human kinome, which consists of approximately 500 kinases, is targeted by clinical candidates [3,4].

The mitogen-activated protein kinase (MAPK) pathway refers to a series of multistep signal transduction pathways involved in the regulation of embryogenesis, cell differentiation, cell growth and programmed cell death, among others [5]. Usually, mitogens (proteins that stimulate cell division), growth factors, osmotic stress, heat shock and proinflammatory cytokines can activate the pathway that involves at least three sets of kinases [6]: a MAP-kinase-kinase-kinase (MAP3K, also MAPKKK), a MAP-kinase-kinase (MAP2K, also MAP-KK, MEK, MKK) and a MAP-kinase (MAPK), which are activated in this series through phosphorylation using adenosine triphosphate (ATP) [5]. While great efforts have been made in drug discovery to target the pathway further downstream like p38 mitogen-activated protein kinase (p38 MAPK), c-Jun N-terminal kinase (JNK), extracellular-signal-regulated kinase (ERK) or further upstream, e.g., Ras, Raf and epidermal growth factor receptor (EGFR), less attention has been paid to the “middle” MKKs. With MKK1 (MEK1) and MKK2 (MEK2), only two of the seven isoforms of the MKK (MAP2K) family have been extensively studied, with several clinical trials underway [7,8,9,10]. However, less effort has been spent on developing inhibitors that target MKK3-7 [11]. The recent entry of MKK4 inhibitor HRX-0215 (structure not disclosed) into clinical trial I (EUDRA-CT No. 2021-000193-28) is the first application under clinical conditions of an inhibitor of the under-explored MKK family [12].

## 2. Classification, Structure and Biological Function of MKK4

Human MKK4, also referred to as MAP2K4, or MAPK/ERK-Kinase 4 (MEK4), JNK kinase 1 (JNKK1), SAPK/ERK kinase-1 (SEK1), consists of 399 amino acids [13,14,15,16]. There are three MKK4 structures publicly available (3aln [17], 3alo [17] and 3vut [18]) displaying that the kinase contains 11 subdomains that fold into a small, N-terminal lobe (composed of five β-sheets and one α-helix) and a larger mainly α-helical C-terminal lobe connected through a flexible hinge region (depicted in Figure 1). The ATP binding site is located in the cleft formed between the two lobes and is surrounded by conserved residues [19]. Notably, the entrance to the ATP binding pocket is open towards the solvent and most likely accessible to the substrate or an inhibitor. MKK4 contains the common MKK-phosphorylation-motif **S**-X-A-K-**T** (Ser257 and Thr261) located in the T-loop of the kinase domain between subdomains VII and VIII [17,18,19,20]. Both hydroxy residues are required to be phosphorylated for full activation of the kinase, as mutation of these residues abolished MKK4’s activity [21]. It has been shown that Ser257 phosphorylation is essential and Thr261 phosphorylation is required for full MKK4 activation [22]. Human non-phosphorylated MKK4 does not show activity [23]. Recent molecular dynamics (MD) simulations suggested that inactive autoinhibited MKK4 exists as a dimer that is destabilized by phosphorylation and therefore activated [24]. MKK4 shares about 50% of its sequence identity with MKK7, the most familiar kinase [19]. Additionally, MKK7 was found to have a similar binding pocket as MKK4, indicating a similar binding mode to small molecule inhibitors [25,26]. Therefore, MKK7 always needs to be considered as off-target. A conserved cysteine (Cys246 in MKK4) upstream of the DFG moiety of all MKKs has been validated for covalent warhead strategies [26,27], as will be discussed in a later section.

During cell response to stress, heat shock, growth factors and proinflammatory cytokines, MKK4 is phosphorylated and activated by the majority of MAP3Ks, including ASK, MEKK, MLK and TAK [28]. While the close relative MKK7 is a specific activator of JNKs, MKK4 can phosphorylate both JNKs and p38 MAPKs [14,15,29,30,31,32]. A recent study has even marked cross-communication between MKK4 and nuclear factor kappa B (NF-κB) pathways by MKK4’s ability to modulate the processing of NF-κB2, and highlighted the possibility to treat NF-κB-caused diseases by MKK4 inhibition [33]. Interactions between MKK4 and MAP3Ks occur via its domain for versatile docking (DVD) at the C-terminus and are mandatory for the activation of MKK4 [34,35]. The docking domain type docking site located at the N-terminal region of MKK4 is responsible for binding its substrates JNK and p38 MAPK [36]. MKK4, together with MKK7, dual phosphorylates the **T**-P-**Y** motif of JNKs and, unlike MKK7, is also able to phosphorylate the **T**-G-**Y** motif of p38 MAPK [37]. While MKK4 prefers the tyrosine residue (Y185) of JNK, MKK7 prefers the threonine (T183) [32,38]. Phosphorylation by MKK7 is essential for JNK activation, while phosphorylation by MKK4 is required for JNK’s full activation [39]. p38 MAPK phosphorylation residues Thr180 and Tyr182 are equally phosphorylated by MKK4, which classifies MKK4 as a dual-specificity protein kinase [39,40]. The roles of p38 MAPK and JNK are pleiotropic and generally involved in processes such as cell proliferation, apoptosis and differentiation [41,42].

In adult mice, MKK4 is ubiquitously expressed [43]. However, in murine embryonic development, the expression pattern differs. Transcription of *MAP2K4* (the gene encoding MKK4) is restricted to the CNS until embryonic day 10 [44]. Starting from embryonic day 10, *MAP2K4* mRNA is found in high levels in the developing liver while the liver cells undergo differentiation up to embryonic day 16 when the transcript levels rapidly drop [44]. *MAP2K4*(−/−) homozygous mice have a high number of proliferating hepatoblasts that cannot differentiate into hepatocytes and undergo apoptosis, leading to death [45], demonstrating MKK4’s significance in embryonic liver cell differentiation.

MKK4, along with MKK7, is crucial for the development of the central and peripheral nervous system (CNS and PNS) as they take part in cell migration, positioning of neuronal cells and commissural fibers development [46,47]. MKK4 appears to be responsible for maintaining the basal activity in neurites and mediates JNK dendritic outgrowth and the creation of neural circuits in the brain [46]. A recent study has demonstrated the impact of the MKK4/MKK7/JNK signaling pathway for controlling the positioning, morphology and differentiation of a hippocampal subpopulation in mice [48]. MKK4, next to MKK7, is important for both retinal development and the injury response following axonal damage, as well as for maintaining retinal ganglion cell survival [49].

There have been conflicting reports on the role of MKK4 in the immune system. Earlier studies attribute a central role to MKK4 for cell survival signaling in T-cell development and proliferation [50,51], whereas new reports observed neither evidence for proliferative defects in MKK4-deficient T-cells nor a negative T-cell response to viral infections [52,53].

**Figure 1 ijms-24-07495-f001:**
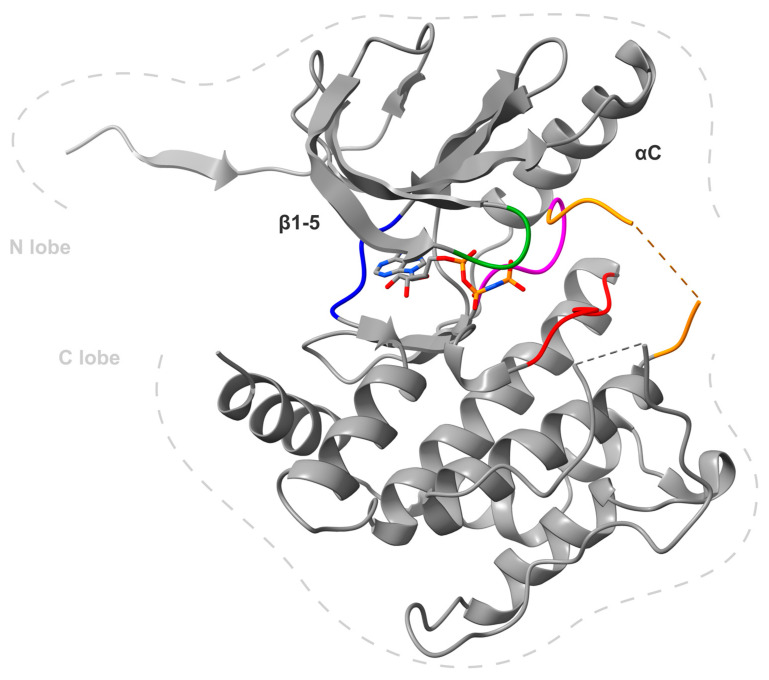
X-ray crystal structure of MKK4 monomer A in a complex with *adenylyl-imidodiphosphate* (PDB: 3aln) [17]. The general structure of protein kinases is clearly visible, with the N-terminal domain at the top (N lobe) dominated by β-sheets and the C-terminal domain at the bottom (C lobe), composed mainly of α-helices. Important structures are colored: hinge region (blue), glycine-rich loop (green), Mg^2+^-binding loop with DFG motif (magenta), catalytic loop with HRD motif (red) and the disordered activation loop, which contains the activation helix but is not shown due to presumably high flexibility (orange). Disordered secondary structures are shown in dashed lines in the protein. Image generated using ChimeraX [54].

## 3. MKK4 Inhibitors

### 3.1. Role of MKK4 Inhibitors in Cancer Therapies

MKK4’s role in tumor development is controversial as it can both act as a tumor suppressor and a tumor promoter. MKK4 is encoded by *MAP2K4* located on chromosomal segment 17p11.2, which can be lost with 7–10% in human epithelial cancers, particularly ovarian and breast cancers [55,56], and was therefore initially presumed to be a tumor suppressor. Additionally, MKK4 has been linked to its suppressing function in pancreatic ductal adenocarcinoma, a cancer type with one of the poorest survival rates of less than 9% [57,58]. Additionally, a human kinome mutation screen of 356 tumors identified *MAP2K4* loss-of-function (LOF) mutations in 11 tumors, supporting MKK4’s suppressive role [59,60,61]. Homozygous deletion of *MAP2K4* is often accompanied by mutations of *TP53* (encoding major tumor suppressor p53) and *KRAS* (encoding common tumor promoter K-Ras) in lung adenocarcinomas [62]. However, tumor-promoting roles of MKK4 have been observed in ovarian, prostate, pancreatic and triple-negative breast cancer [43,63,64,65,66]. MKK4 promotes prostate and ovarian cancer metastasis [66,67]. Prostate cancer is the most commonly diagnosed cancer type in men in the United States (268,490 estimated new cases in 2022) and the second most common form of cancer death (34,500 estimated deaths in 2022), comprising great significance [68,69]. While the 5-year relative survival rate is well above 99% for localized and regional prostate cancer, the survival rate dramatically drops to 32% for advanced cancer stages [69]. MKK4 has been validated as a target for prostate cancer given the fact that MKK4 inhibition could prevent cell invasion and metastasis in preclinical studies [67,70,71]. Additionally, it has been demonstrated that microRNAs directly targeting MKK4 are downregulated in prostate cancer tissues and cell lines, making MKK4 a promising target for prostate cancer therapy [72]. Furthermore, ultraviolet B (UVB) radiation was shown to activate p38 MAPK and JNK pathways and it can lead to skin cancer [73]. Hence, MKK4 inhibition effectively shuts down these pathways [74]. Moreover, the concurrent inhibition of MKK4/JNK and PI3K/Akt pathways in non-small cell lung cancer line H1299 showed antiproliferative effects [75].

Taken together, depending on the genetic setting of the tumor, MKK4 can act as tumor-suppressive or tumor-promoting, a common characteristic of many kinases [76]. In particular, high-impact tumors with low survival rates, such as pancreatic, triple-negative breast cancer and metastatic prostate cancer, make MKK4 an interesting target for small molecules. To date, only a few MKK4 inhibitors have been developed. In the beginning, mostly natural products with protective effects were studied and this effect was attributed to MKK4 suppression. Only recently, modern medicinal chemistry methods have been applied to develop selective and active structures for MKK4. An overview of all known structures is described below.

#### 3.1.1. 9-H-pyrimido [4,5-b]ind-6-ol-scaffold

In 2003/2004, Bayer published a dual MKK4 and MKK7 inhibitor with a 9-H-pyrimido [4,5-b]ind-6-ol-based scaffold [77,78]. Based on the fact that MKK7 and MKK4 are the only two known kinases of the MKK family that activate JNK, and MKK4 also phosphorylates p38 MAPK [28,79], the project focused on the development of a specific MKK7 and/or MKK4 inhibitor. The objective was to interfere with the synthesis of pro-inflammatory cytokines and the activation of several immune cells to target inflammatory and immunoregulatory diseases on the one hand, but also to get involved in JNK activation and its role in tumor cell survival [64,80]. Compound **1** (Figure 2) showed the highest activity. An IC_50_ value of <1 µM was reported for both MKK4 and MKK7.

#### 3.1.2. 7,3′,4′-Trihydroxyisoflavone (THIF)

7,3′,4′-THIF (**2,** Figure 3) is a major metabolite of Daidzein (**3**, Figure 3), a natural isoflavone found in certain beans, legumes and sprouts [81,82]. Isoflavones [83] and soy extracts [84] have been reported to have a photoprotective effect in skin tissue and, therefore, both components were investigated for their effect on ultraviolet B (UVB)-induced skin cancer. At first, 7,3′,4′-THIF and Daidzein were tested for their inhibitory effect on UVB-induced cyclooxygenase 2 (COX-2) expression in JB6 P+ cells (murine skin cells). While 7,3′,4′-THIF had an inhibitory effect on the expression, Daidzein had no effect. In mouse skin models, a lack of COX-2 was identified as protection against UVB-induced skin cancer while overexpression led to tumor-promoting activity [85]. Furthermore, 7,3′,4′-THIF significantly reduces the NF-κB transcription activity which is part of the down-regulation of UVB-induced COX-2 expression [86]. In comparison, it was also more effective than Daidzein. Moreover, data revealed that 7,3′,4′-THIF suppresses the UVB-induced phosphorylation of JNK and p38 MAPK in JB6 P+ cells. As JNK and p38 MAPK are both components of a stress-activated protein kinase signaling pathway [13], the upstream regulatory proteins were further investigated. It was verified that 7,3′,4′-THIF inhibited MKK4 and Cot (MAP3K8), a kinase that guides the activation of JNK, NF-κB and p38 MAPK. In vitro kinase assays unveiled the inhibition of MKK4 and Cot and demonstrated no effect on either MKK3 and MKK6 or p38α MAPK and JNK. These results led to the assumption that the effect on the COX-2 expression was caused mostly by the ATP-competitive MKK4 and Cot inhibition through 7,3′,4′-THIF. Therefore, it underlines the potential of MKK4 inhibition in the treatment of UVB-induced skin cancer [74].

#### 3.1.3. Genistein

Genistein (**4,**
Figure 4) is the major isoflavone in soybeans and was investigated for its potential for cancer treatment by MKK4 inhibition [70,87]. The consumption of Genistein through soy is associated with lower rates of metastatic prostate cancer [88,89,90]. Therefore, the steps of the metastatic cascade were examined more closely. The cell invasion is the initial, necessary step in cancer metastasis, and transforming growth factor-*β* (TGF-*β*) was shown to increase it. For a better understanding, the downstream processes of TGF-*β* were further investigated. TGF-*β* activates MKK4, which upon activation phosphorylates downstream effector proteins, including p38 MAPK. p38 MAPK phosphorylates MAPK-activated protein kinase 2 (MAPKAPK2), which itself phosphorylates heat shock protein 27 (HSP27). This finally leads to the expression of matrix metalloproteinase-2 (MMP-2) and the increase in cell invasion [91,92]. In further investigations, MKK4 was found to be the target for Genistein in the treatment of prostate cancer cells with an IC_50_ value of 400 nM. In a phase II clinical study, higher expression of MKK4 was associated with higher MMP-2 expression and cell invasion, giving evidence for the anti-invasive activity of Genistein in prostate cancer treatment. Ultimately, the clinical study had limitations and it could not be proven that Genistein binds to MKK4’s active site, even though it was predicted through computer-simulated structural modeling that Genistein directly inhibits MKK4 activity [70]. However, Genistein has been studied extensively, especially for its role as a cancer therapeutic, with 75 studies to date investigating or including Genistein (according to clinicaltrials.gov), along with a phase II clinical trial investigating Genistein treatment for metastatic prostate cancer [93]. It has been identified as an inhibitor for multiple targets, such as the epidermal growth factor receptor (EGFR) [94,95], cyclin-dependent kinases (CDK) [96,97], polo-like kinase 1 (PLK1) [98,99,100] and estrogen receptor (ER) [101].

#### 3.1.4. Dehydroglyasperin C

Dehydroglyasperin C (DGC) (**5**, Figure 5) is a natural product that raised awareness for being the major flavonoid compound in ethanolic licorice extract [102]. There are many reports on the bioactivity of licorice extract, which include antioxidant, anti-inflammatory, anti-carcinogenic, hepatoprotective and antimicrobial activity [103,104,105,106,107,108]. Mechanistic studies of DGC revealed that the inhibition of 12-O-tetradecanoylphorbol-13-acetate (TPA)—a potent tumor promoter [109,110]—induced phosphorylation of p38 MAPK, JNK and Akt, a major substrate of phosphoinositide 3-kinase (PI3K) [111]. A pull-down assay with DGC-conjugated sepharose beads disclosed direct physical binding of DGC to the upstream regulators MKK4 and PI3K. ATP titration indicated an ATP-competitive mechanism of inhibition of DGC to both kinases [111]. DGC was able to inhibit tumor promoter-induced neoplastic transformation in cells and could also suppress UVB-induced COX-2 expression by inhibiting the MKK4 and PI3K pathways. The authors highlighted the importance of multi-target kinase inhibitor approaches to cancer therapies, as highly selective inhibitors are thought to contribute to drug resistance in cancer patients [111].

#### 3.1.5. HWY336

The protoberberine derivative HWY336 (**6**, Figure 6) was identified as a dual MKK4 and MKK7 inhibitor in a library screen of 80 protoberberine derivatives. IC_50_ values of 6 µM on MKK4 and 10 µM on MKK7 in vitro were reported. Protoberberine alkaloids are present in many plant families, including poppies, barberries and laurels [112], and can show anti-tumor activity in vivo [113,114]. Therefore, a library screen was performed with MKK1, MKK2, MKK3, MKK4, MKK6, MKK7 and MAPKs (JNKs, p38 MAPKs, ERKs), finding HWY336 as a selective dual MKK4/MKK7 inhibitor [115].

Subsequently, human embryonic kidney 293 (HEK293T) cells that were treated with D-sorbitol, to activate JNK, were incubated with HWY336 in increasing concentrations. The phosphorylation of JNKs was reduced after 3 h of incubation with 9 µM HWY336 and disappeared after 4 h of treatment with 12 µM HWY336. These experiments and the performed wash experiments indicate that the inhibition is directed to MKK4, is reversible and is non-covalent. 

For a better understanding of the inhibitory selectivity, molecular docking and phylogenetic analysis based on sequence homology was performed. The structural model supported the hypothesis that HWY336 is a non-ATP competitive inhibitor of MKK4/MKK7. It also led to the assumption that HWY336 inhibits the phosphorylation of MKK4/MKK7 inside the activation loop or hinders the substrate to access the kinase [115].

#### 3.1.6. 3-Arylindazoles

Due to MKK4’s role in prostate cancer metastasis, it has become a target of interest for therapeutic inhibition. To discover new and especially selective inhibitors, a high-throughput screen with a library of 50,000 diverse compounds was carried out. Hereafter, the work was focused on a small hit compound with an indazole core (**7**, Figure 7) with an IC_50_ value of 190 nM towards MKK4 [116]. After the confirmation of the ATP-competitive mechanism of inhibition through ATP titration and a fluorescence thermal shift (FTS) assay, which neither showed a preference for the binding of the active or the inactive forms of MKK4, the MKK4 selective inhibition within the MKK family was investigated. **7** has a 30- to 60-fold selectivity to MKK1 with an IC_50_ value of 12 µM which marks the highest inhibition in the MKK family after MKK4. The modeling of **7** with MKK4 led to predictions of key interactions. Hydrogen bond interactions in the hinge region with the residues Leu180 and Met181, as well as electrostatic interaction of the carboxylate of **7** and Lys187, were observed. Furthermore, there appeared to be a hydrophobic back pocket that could be occupied starting from the 5- and 6- positions of the indazole core. Based on this, 6-fluoro-3-arylindazole (**8**, Figure 7) was obtained and showed selectivity within the MKK family and IC_50_ values of 41 nM. Subsequently, a kinase screen with another 50 kinases was performed at 10 µM. **13** inhibited 12 of the 57 kinases, while **7** only inhibited 5 kinases [116]. Further investigation of the temporary lead structure **8** was carried out with two different approaches: the synthesis of noncovalent and covalent inhibitors.

For the noncovalent approach, replacing the carboxylic acid of **8** with the bioisosteric sulfonamide gave compound **9** (Figure 7) with improved activity and membrane permeability. The primary para-substituted sulfonamide yielded an IC_50_ value of 61 nM. Based on **9**, various analogs were investigated and led to **10** (Figure 7) with an IC_50_ value of 83 nM. The selectivity of **9** and **10** were examined. **9** and **10** were first tested in their selectivity profile within the MKK family. Even though **9** proved a better selectivity profile in the MKK family, **10** was tested against 97 kinases at a concentration of 10 µM. Out of 97 of these kinases, 28 were hit by at least 80% inhibition, including tyrosin-like kinases (TK), tyrosine kinase-like kinases (TKL), serine/threonine kinases (STE), cyclin-dependent kinases, mitogen-activated protein kinases, glycogen synthase kinases and CDK-like kinases [117]. In other studies, the co-treatment of **9** and **10** with an MKK1/2 inhibitor (U0126) in pancreatic cancer cell proliferation was examined. 

Computational modeling also suggested a covalent approach and several proposed covalent inhibitors were generated. Using **8** as a starting point, different electrophilic warheads were added to the ortho-position of the aryl ring while the carboxylic acid was removed. However, none of the compounds were able to bind covalently and did not show activity in the noncovalent control.

#### 3.1.7. BSJ-04-122

Inhibition of MKK4 and MKK7 was investigated in the triple-negative breast cancer line MDA-MB-231 as the genetic knockdown of MKK4 in MDA-MB-231 resulted in a suppressed tumor growth in a mouse xenograft model [63]. The kinase inhibitor SM1-71 (**11**, Figure 8) was used to target a conserved cysteine located before the DFG motif that can be found across the whole MKK family [27]. To improve kinase selectivity, further analogs of **11** were synthesized, resulting in BSJ-04-122 (**12**, Figure 8) with an IC_50_ value of 4 nM on MKK4. The inhibition of **12** was also tested for MKK7 activity using two different assays, resulting in sub-micromolar IC_50_ values, while a reversible saturated analog (not shown) showed values above 10 µM. The covalent binding was confirmed with liquid chromatography-mass spectrometry (LC-MS) showing a molecular weight shift according to the adduct of MKK4 and **12**. Chymotrypsin digestion followed by MS analysis of the protein fragments revealed that Cys247 in MKK4 and Cys261 in MKK7 were covalently connected to BSJ-04.122. Upon treatment of cells with **12**, a competition-based pulldown assay showed that kinases MKK1/2/3/5/6 were not engaged, which suggested high selectivity among the MKK family. Docking experiments of **12** with MKK4 (3aln [17]) predicted the formation of two hinge hydrogen bonds through pyrrolopyrimidine with Met181 and Glu170 and an additional hydrogen bond between the acrylamide and the Ser233 backbone carbonyl. The conformation of 3-aminophenyl acrylamide is in proximity to Cys246 for covalent binding, while docking with MKK2 (1S9I [118]), despite its high similarity with MKK4, shows an extended hinge with an inserted extra residue, Glu152. Additionally, MKK2 contains Asp151 and His149, which are charged, while MKK4 inherits neutral residues Leu180 and Ser182. Therefore, not only are the hydrophobic interactions of **12** with MKK4 stronger, but Lys187 also forms a cation–π interaction with the phenyl ring. MKK7 was found to have a similar ATP-binding pocket to MKK4, indicating a similar binding mode. Ultimately, **12** was investigated to determine whether it induces cell growth inhibition of triple-negative breast cancer cell lines. Surprisingly, concentrations ranging from 4 nM to 10 µM did not affect the proliferation rate. In a combinatorial therapy approach, a combination of **12** and JNK inhibitor JNK-IN-8 demonstrated an enhanced antiproliferative effect in cells [26].

### 3.2. Role of MKK4 Inhibitors in Liver Regeneration

In the previous section, MKK4’s significance in the differentiation of early-stage hepatocytes for the development of embryonic livers was described. The abundance of *MAP2K4* resulted in rapid apoptosis of the primordial liver and ultimately led to the death of mice embryos. Contrariwise, an in vivo RNAi screen has identified MKK4 as a key enzyme in liver regeneration [119]. As illustrated in Figure 9, MKK4 silencing by shRNA led to increased signaling through apoptosis signal-regulating kinase 1 (ASK1), a MAP3 kinase and MKK7, thus yielding higher phosphorylation of JNK1. Thereby, through the phosphorylation of ETS transcription factor 1 (ELK1) and the activation of transcription factor 2 (ATF2), the substrates of JNK1 are intensified, which resulted in faster cell-cycle entry and progression of hepatocytes during liver regeneration [119]. Moreover, resting, non-regenerating livers with MKK4 silencing were shown to be resistant to Fas-induced apoptosis. Although increased proliferation and resistance to apoptosis are characteristics of cancer, in mice with stable intrahepatic MKK4-knockdown, no tumors could be observed, indicating that the absence of MKK4 is not a strong tumor-initiating factor and making MKK4 a promising target for transient pharmacological inhibition and a liver regeneration therapeutic [119]. 

In the EU and U.S., more than 250,000 patients per year suffer from acute liver disease and 700,000 patients suffer from chronic liver disease [120]. There are approximately 2 million or 3.5% of worldwide deaths per year, with hepatocellular carcinoma being the most common primary liver cancer, which is considered the fifth most common cancer and is the third leading cause of cancer-related death in the U.S. [121]. With liver transplantation as the only curative option for the treatment of acute liver failure and end-stage chronic liver failure, only a limited number of donor livers are available and the need for new therapies such as MKK4 inhibition is eminent [119]. MKK7 and JNK1 are to be considered anti-targets. Since their activity is responsible for liver regeneration [119,122], these must not be targeted by the inhibitors. In recent years, the development of selective MKK4 inhibitors serving liver therapy purposes has risen [119,123,124,125].

#### 3.2.1. Azaindoles

Following a target-hopping strategy from FDA-approved B-Raf^V600E^ (serine/threonine-protein kinase B-Raf) inhibitor Vemurafenib, which showed off-target activity towards MKK4, a multi-parameter optimization process emphasizing a distinct SAR (structure–activity relationship) was performed, leading to compounds **13**, **14** and **15** (Figure 10). To determine affinities towards MKK4 and off-targets, commercially available KINOMEscan technology was used, and binding affinities are given as percentage of control (POC) values. In the assay, the compounds compete with a proprietary immobilized multikinase inhibitor for the binding site of the DNA-tagged kinase. The readout is via quantitative PCR and the POC is calculated from the DMSO (dimethyl sulfoxide) control [126]. A POC of 0 is equal to a complete displacement of the ligand from the kinase (complete binding) and a POC of 100 is equal to no binding of the compound to the kinase. Starting from Vemurafenib (POC^MKK4^ = 14 @ 0.1 µM; POC^B-Raf^ = 16 @ 0.1 µM), this demonstrated the design of new inhibitors with a high affinity to MKK4 (POC^MKK4^ = 2.2, 0.2 and 0.35 @ 0.1 µM), with selectivity over the off-targets MKK7, JNK1, B-Raf, MAP4K5 and ZAK in the range of factors 40 to 190, being the first-in-class inhibitors targeting MKK4 for hepatocyte proliferation [123].

#### 3.2.2. Pyrazolopyridines

Further exploration of the core structure led to changing the azaindole to pyrazolopyridine. SAR experiments resulted in compound **16** (Figure 11). It contains an acetylated sulfonamide instead of Vemurafenib chlorine. It showed the highest activity towards MKK4 (IC_50_ = 29 nM, POC^MKK4^ = 0.35 @ 100 nM). However, the selectivity was poor towards B-Raf, showing even higher activity (POC^B-Raf^ = 0 @ 0.1 µM). In compound **17**, the alkyl side chain was replaced by a benzyl, dramatically shifting the selectivity from B-Raf (POC^B-Raf^ = 76 @ 0.1 µM) to MKK4 (POC^MKK4^ = 0.4 @ 0.1 µM) with an IC_50_ value of 146 nM. K_D_ values of **17** showed remarkable selectivity towards the off- and anti-targets by factors of >100, >250, >600 and almost 3000 against JNK1, B-Raf, MAP4K5, MKK7 and ZAK, respectively. **17** was screened against a panel of 97 kinases at 1 µM, a concentration that is 500 times higher than **17**s Kd value for MKK4. Only two kinases (AURKB and SNARK) showed a POC value of <35%, demonstrating the high kinome selectivity of **17** [124].

#### 3.2.3. Carbolines

By replacing the azaindole/pyrazolopyridine scaffold with α-carboline as a new scaffold, highly potent inhibitors with an extraordinary and robust selectivity profile driven by the intrinsic selectivity of the α-carboline were obtained [125]. Activity towards the previously named off-targets could be minimized to a non-relevant level making this type of inhibitor unique in the class of known MKK4 inhibitors, as it allows a broad range of activity-increasing and pharmacokinetic-improving modifications without detrimental selectivity effects. Modifications on the azaindole/pyrazolopyridine scaffold had to balance potency and selectivity well. The increased steric demand and its rigidized structure proved to be a very robust selectivity-inducing structural element without affecting potency. All carboline-containing compounds showed good activity and selectivity. **18** (Figure 12A) was selected to be screened against 320 kinases at a concentration of 100 nM resulting in a selectivity factor of S(20) = 0.103 [125,127]. A further study based on this new scaffold was recently published, dealing with a fluorescence-labeled carboline derivative. We were able to detain a linker attached to a bulky fluorophore while retaining MKK4 affinity and selectivity (Figure 12B) [128]. Since the available MKK4 crystal structures do not consider the lipophilic back pocket, docking into our molecular dynamics (MD)-simulated structure of MKK4 [24] was crucial for the success of the design and implementation of the fluorophore. The binding mode was similar to the related structure of Vemurafenib in the crystal structure of its original target B-Raf (4rzv, 3og7) [129], where the azaindole nitrogens form two hydrogen bonds to the hinge region, while the *para*-chlorophenyl protrudes from the ATP-binding pocket into the solvent-exposed region. The NH of the sulfonamide interacts with the DFG motif of B-Raf and forms hydrogen bonds with the aspartate, while the isopropyl chain occupies the lipophilic back pocket. Therefore, chlorophenyl was used for detaining the linker-connected fluorophore, providing a potent fluorescent ligand for competitive high-throughput screening [128].

### 3.3. Role of MKK4 Inhibitors in Other Diseases

To date, little is known about MKK4’s influence on other diseases, but reports have risen in the past few years. MKK4 is a negative regulator of the TGF-*β*_1_ (transforming growth factor-*β*_1_) signaling pathway associated with remodeling and arrhythmogenesis with age, and has been suggested as a potential therapeutic target for the treatment of atrial fibrillation [130]. It has been demonstrated that a specific microRNA directly targeting MKK4 is downregulated in osteoarthritis development. Therefore, MKK4 levels were increased in osteoarthritis cartilage leading to downstream activation of matrix-degrading enzymes, making it a promising target for osteoarthritis therapy [131]. MKK4 was investigated in neuroprotection studies [132] and has been implicated in neurological conditions, including stroke, Parkinson’s, Huntington’s and Alzheimer’s disease [132]. In addition, it was possible to show that MKK4 suppression rescues neuronal cells from cell death using prenylated quinoline carboxylic acids (PQA) [133].

#### Prenylated Quinoline Carboxylic Acids

Ppc-1 (**19**, Figure 13) is a secondary metabolite synthesized by a cellular slime mold and acts as a neuroprotective agent on glutamate-induced cell death in hippocampal cell cultures with an IC_50_ value of 239 nM [133]. Further derivatization by the authors led to compound PQA-11 (**20**, Figure 13) with a saturated isopentyl chain, enhancing activity with an IC_50_ value of 127 nM. In addition, PQA-11 inhibited neurotoxin-induced cell death. MKK4 was identified as the target of PQA-11 being responsible for JNK activation and ultimately caspase-3 activation, leading to diminished apoptosis. The effect was confirmed with MKK4 siRNA in vitro. The quartz crystal microbalance (QCM) method was used to verify the direct interaction of PQA-11 with MKK4 in a concentration-dependent manner. Notably, no interaction was observed with a constitutively active type of MKK4, suggesting the binding of PQA-11 to inactive MKK4. It has been proposed that sphingosine is involved in JNK pathway activation [134]. Studying the interaction of sphingosine on MKK4 revealed that it induced autophosphorylation of MKK4, which is inhibited by PQA-11 in a concentration-dependent manner. Through QCM-, assay- and docking-based data, the authors present a new mode of inhibition, in which the inhibitor PQA-11 displaces sphingosine from a binding pocket competitively. Since MKK4 requires sphingosine binding for autophosphorylation, a reduction in activity can be achieved by using PQA-11 as a lead structure for the further development of neuroprotective drugs and the treatment of neurodegenerative diseases such as Alzheimer’s or Parkinson’s disease.

## 4. Conclusions

MKK4 is a dual-specificity kinase that regulates cellular signaling pathways by activating JNK and p38 MAPK. Recently, even a cross-communication between MKK4 and NFκB was reported, opening the possibility to tune this pathway and related pathologies by MKK4 inhibition.

Research into the function of MKK4 and diseases associated with it has led to increasing interest in drug development, especially in recent years. Previous approaches used natural products as active ingredients with activity on MKK4 to study and validate the kinase as a pharmacological target. Thus, MKK4 was identified not only in certain cancer types but also as being responsible for transformation into cell invasion and metastasis in high-impact cancer types, such as prostate and ovarian cancer. This laid the foundation for the targeted development of selective MKK4 inhibitors, which quickly achieved success with nanomolar inhibitory small molecules that bind not only reversibly but also covalently. However, more challenges lie ahead in the field of cancer therapy. The effectiveness of the inhibitors still needs to be demonstrated in suitable in vivo models.

The identification of MKK4 as a key regulator in liver regeneration has generated great interest as it offers a solution to the current lack of alternatives to liver therapies such as transplantation. Notably, unlike in cancer therapies, where inhibitors are often developed as dual-specific against MKK4 and MKK7, high selectivity towards MKK7 is essential in liver regeneration. Recently, several structures serving this purpose have been published and the entry of inhibitor HRX-0215 (structure not disclosed) into clinical trials marked a milestone in the drug development of MKK4, as it is the first clinically studied inhibitor to target one of the “non-classical” MKKs (MKK3-7).

MKK4 has also been shown to be a target in other conditions, including osteoarthritis, atrial fibrillation and neurodegenerative diseases such as Alzheimer’s and Parkinson’s disease. It has been shown to prevent cells from apoptosis, autodegradation and senescence. Although studies on in vivo models are still required to prove the efficacy, MKK4 inhibitors show great potential to be used as therapy, which highlights the need for novel MKK4 therapeutic agents.

In summary, MKK4 has been validated for certain cancer diseases and liver regeneration therapies. While dual-inhibitor specificity towards MKK7 can be beneficial for a complete MKK4/MKK7/JNK pathway shut down in cancer cells, high MKK4 selectivity is mandatory in liver regeneration. While the first small molecules inhibited MKK4 only nonselectively in the μM range, mostly originating from natural products, several highly selective inhibitors, effective at nanomolar concentrations, have recently been published that even involve covalent strategies and have a clinical candidate in the pipeline.

However, there are still challenges ahead. These include further investigation of the role of MKK4 in other diseases, especially in neuroprotection, and a possible application of an MKK4 inhibitor for neurodegenerative diseases. So far, the binding modes of inhibitors have been established only by homology models. For rational drug design, a crystal structure that represents the binding mode at high resolution would be a breakthrough. Most structures are thought to be ATP-competitive type 1 kinase inhibitors. Since the ATP binding pockets are often conserved, obtaining selectivity is problematic. In addition, these inhibitors must overcome high endogenous ATP levels to be effective intracellularly. Accordingly, covalent or allosteric inhibitors are promising options, which have also been discussed in previous sections. Particularly, resolving the molecular mechanism of the latter would help transform lead structures into drug candidates.

## Figures and Tables

**Figure 2 ijms-24-07495-f002:**
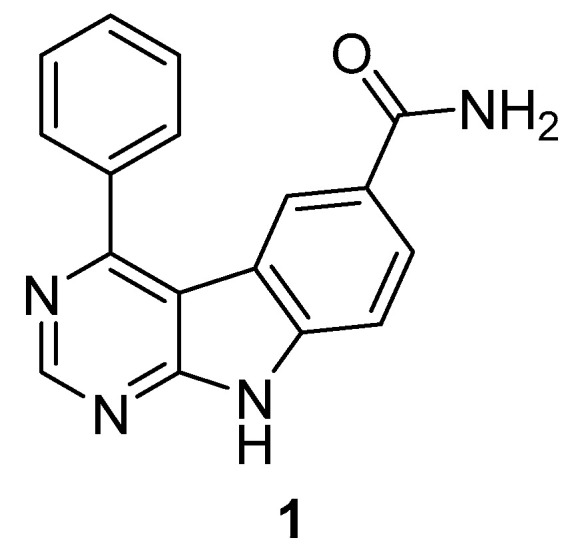
Most potent 9-H-pyrimido [4,5-b]ind-6-ol-based structure.

**Figure 3 ijms-24-07495-f003:**
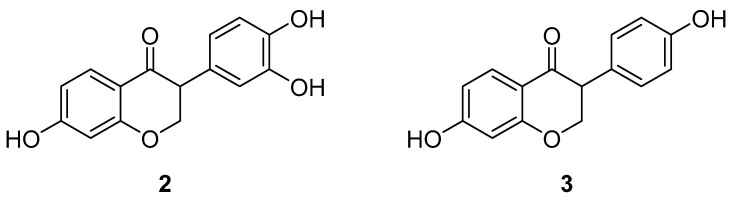
7,3′,4′-Trihydroxyisoflavone (**2**) and Daidzein (**3**).

**Figure 4 ijms-24-07495-f004:**
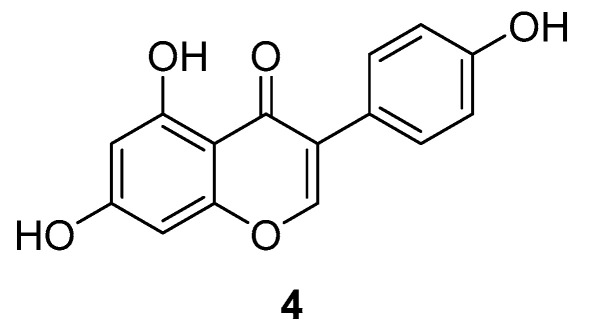
Genistein.

**Figure 5 ijms-24-07495-f005:**
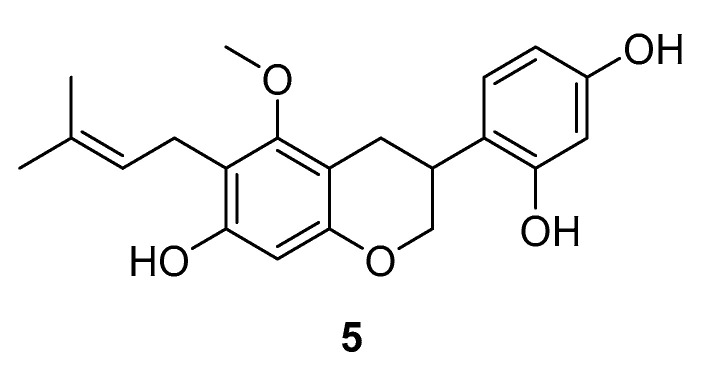
Dehydroglyasperin (**5**).

**Figure 6 ijms-24-07495-f006:**
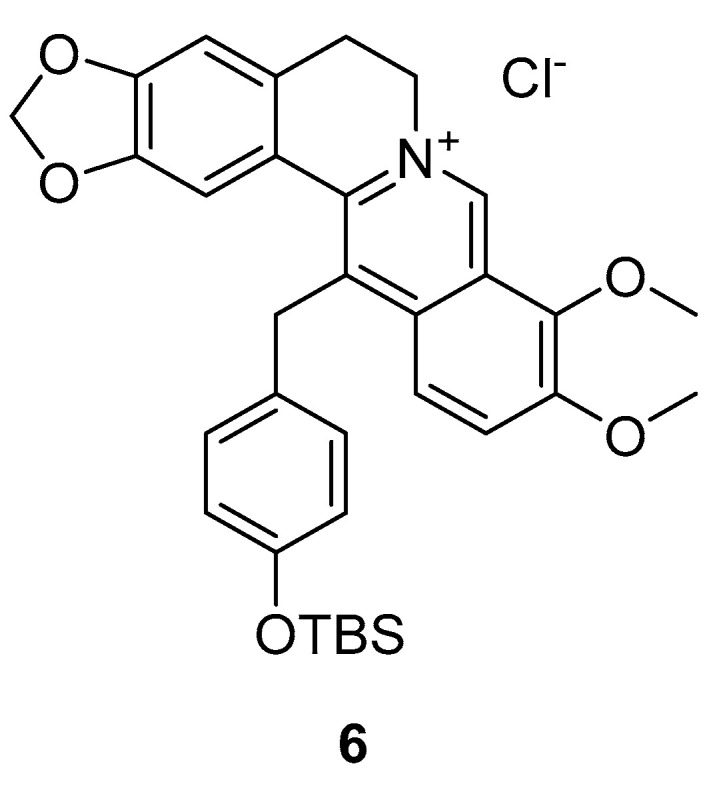
HWY336.

**Figure 7 ijms-24-07495-f007:**
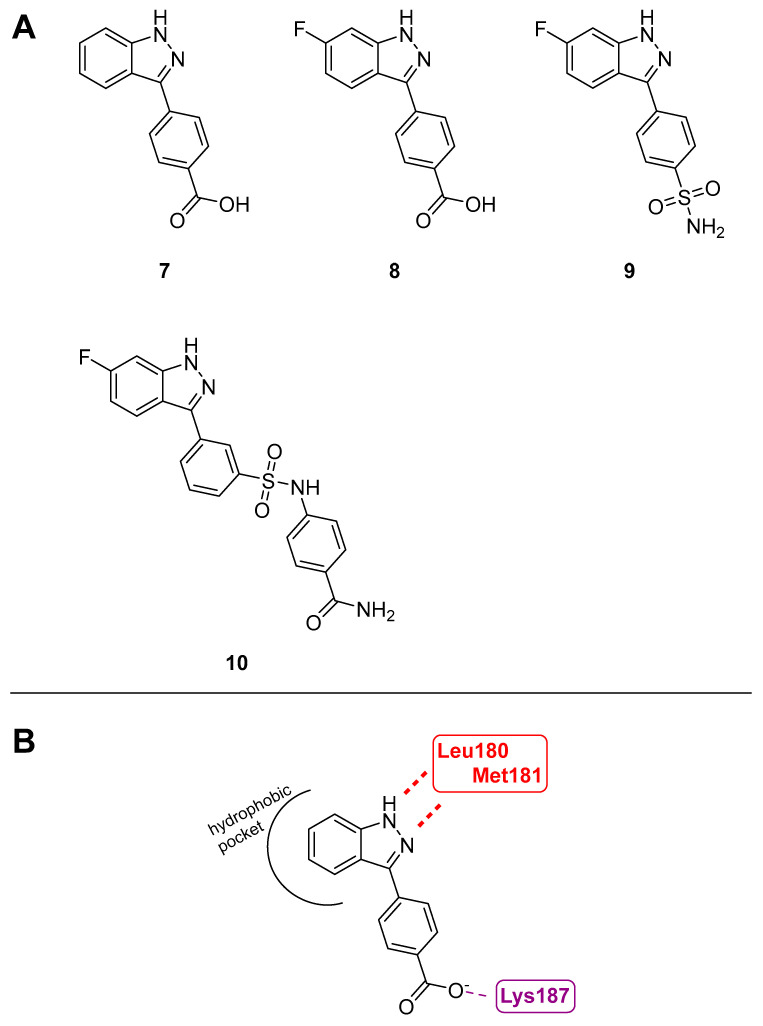
(**A**): 3-Arylindazoles; (**B**): predicted binding pose of 7 in MKK4 (3aln) according to Deibler et al. [25].

**Figure 8 ijms-24-07495-f008:**
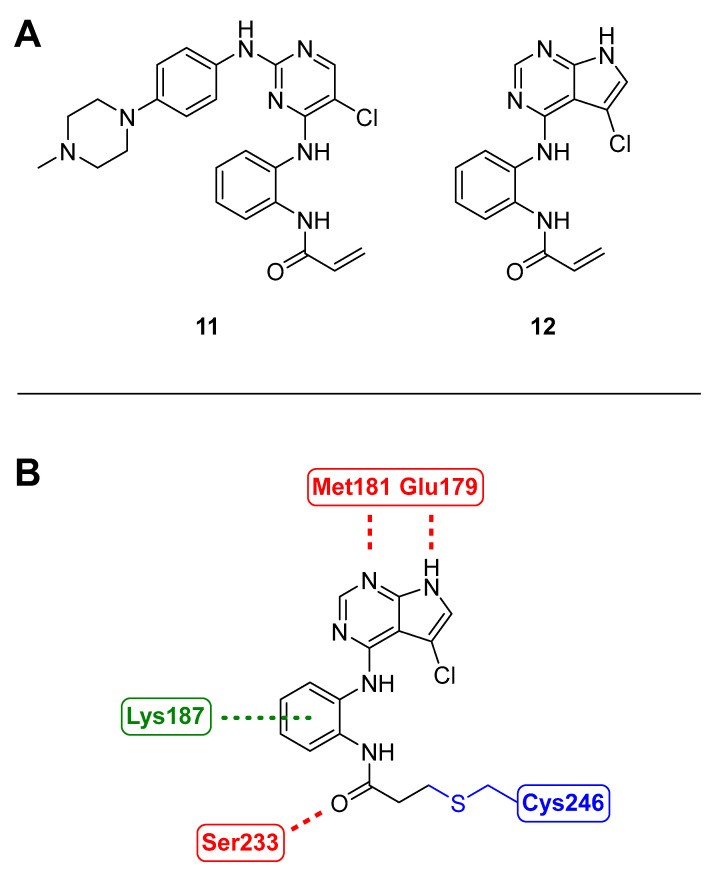
(**A**): SM1-71 (**11**) and BSJ-04-122 (**12**); (**B**): covalently docked BSJ-04-122 to MKK4 (3aln) according to Jiang et al. [26].

**Figure 9 ijms-24-07495-f009:**
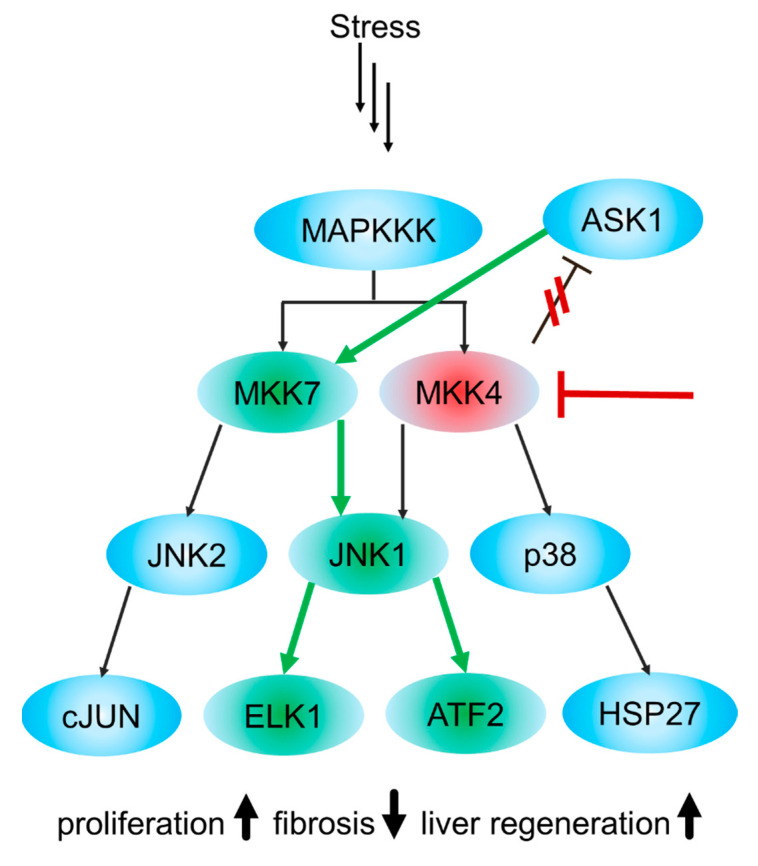
Overview of the inhibited MKK4 signaling pathway. Red arrow shows the downregulation or inhibition of MKK4. Red lines demonstrate the lack of negative feedback effect of MKK4 on ASK1. Green arrows show increased activation of the MAP kinase cascade due to an absence of MKK4 activity. Adapted from Wuestefeld et al. [119].

**Figure 10 ijms-24-07495-f010:**
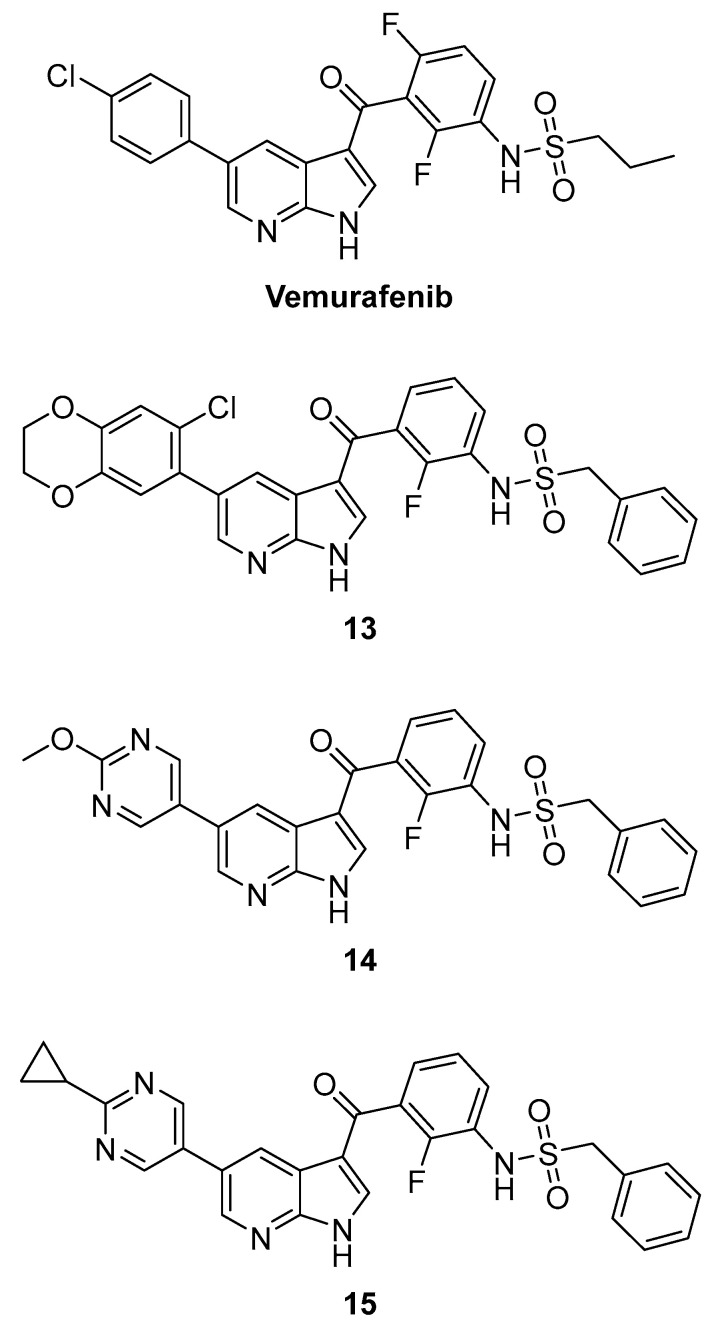
Azaindoles.

**Figure 11 ijms-24-07495-f011:**
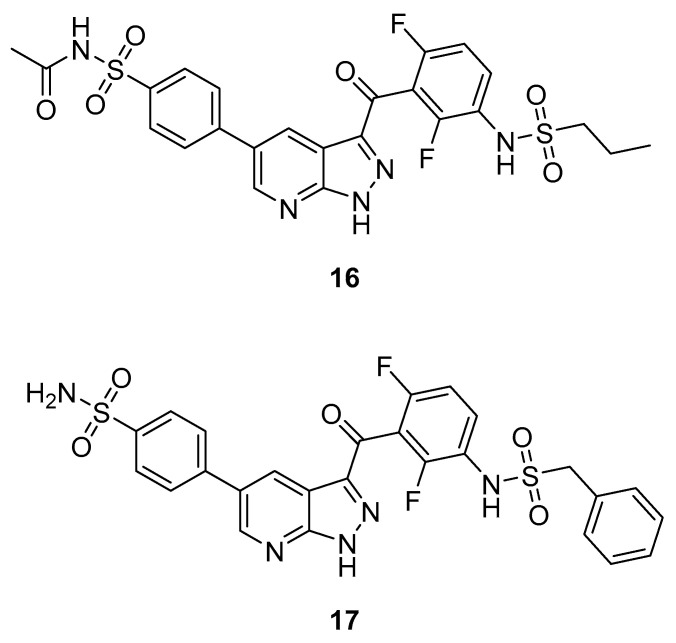
Pyrazolopyridines.

**Figure 12 ijms-24-07495-f012:**
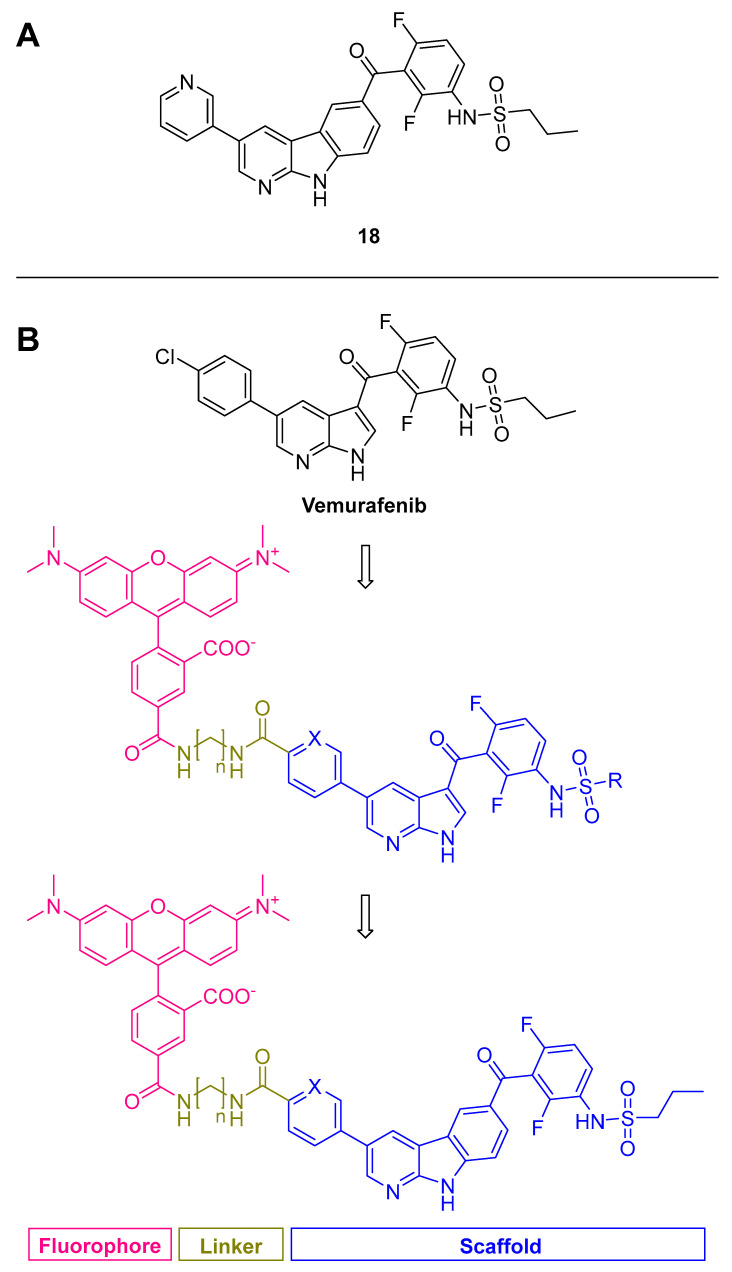
(**A**): Carboline compound **18** that was selected for screening. (**B**): Overview of the fluorophore attachment strategy. Scaffold with X = CH or N and R = propyl or benzyl (blue), attached by a linker with n = 1–6 (gold) to the fluorophore 5-TAMRA (5-carboxytetramethylrhodamine; pink). Adapted from Kircher et al. [128].

**Figure 13 ijms-24-07495-f013:**
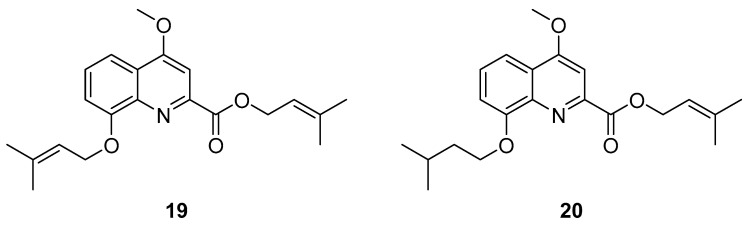
Ppc-1 (**19**) and PQA-11 (**20**).

## Data Availability

Data is contained within the article.

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
