# Peer review of "MKK4 Inhibitors—Recent Development Status and Therapeutic Potential"

_ijms, 2023, doi:10.3390/ijms24087495_

Round 1

Reviewer 1 Report

In the first part of this manuscript, authors summarize the information on mitogen-activated protein kinase kinase 4 (MKK4) and give notions that MKK4 is a promising target for treatment of cancer, liver-associated diseases, and others, and that MKK4 inhibitors might be used as effective drugs. In the following main part, they describe structures, specificity, potency, and action mechanism of various MKK4 inhibitors in detail and refer to future challenges for actual clinical use of MKK4 inhibitors. This manuscript thus contains useful information for future study and use of MKK4 inhibitors.

Comment 1:

“Instructions for Authors” requires the following: Acronyms/Abbreviations/Initialisms should be defined the first time they appear in each of three sections: the abstract; the main text; the first figure or table. However, this instruction is not always followed in this manuscript. In addition, several abbreviations are used for one molecule or enzyme (e.g., ‘p38’, ‘p38 MAPK’, and ‘p38 MAP kinase’ are used for ‘p38 mitogen-activated protein kinase’). Some revision on the use of abbreviations would therefore further enhance the readability of the manuscript.

Comment 2:

On p. 1, lines 13-14, there is the following description: MKK4 (or MEK4) is a dual-specific protein kinase that phosphorylates and regulates both JNK and p38 MAPK signaling pathways … . Similar description is on p. 15, lines 461-462. According to my understanding, however, it is likely that ‘dual-specificity protein kinase’ is more often used than ‘dual-specific protein kinase’, and that a dual-specificity protein kinase is an enzyme catalyzing phosphorylation of both tyrosine and serine/threonine residues. The above description in the manuscript might be somewhat misleading. Please confirm the accuracy of the description.

Comment 3:

On p. 15, lines 426-427, ‘tumor growth factor-b1’ should be ‘transforming growth factor-b1’.

Comment 4:

According to “Instructions for Authors”, the abstract should be a single paragraph.

Author Response

Comment 1:

“Instructions for Authors” requires the following: Acronyms/Abbreviations/Initialisms should be defined the first time they appear in each of three sections: the abstract; the main text; the first figure or table. However, this instruction is not always followed in this manuscript. In addition, several abbreviations are used for one molecule or enzyme (e.g., ‘p38’, ‘p38 MAPK’, and ‘p38 MAP kinase’ are used for ‘p38 mitogen-activated protein kinase’). Some revision on the use of abbreviations would therefore further enhance the readability of the manuscript.

Response 1: We agree it is confusing. In the updated version, we defined every abbreviation in the text and used p38 MAPK uniformly.

Comment 2:

On p. 1, lines 13-14, there is the following description: MKK4 (or MEK4) is a dual-specific protein kinase that phosphorylates and regulates both JNK and p38 MAPK signaling pathways … . Similar description is on p. 15, lines 461-462. According to my understanding, however, it is likely that ‘dual-specificity protein kinase’ is more often used than ‘dual-specific protein kinase’, and that a dual-specificity protein kinase is an enzyme catalyzing phosphorylation of both tyrosine and serine/threonine residues. The above description in the manuscript might be somewhat misleading. Please confirm the accuracy of the description.

Response 2: In the updated version, we changed "dual-specific" to "dual-specificity" and to avoid misleading information explained what classifies MKK4 as a dual specificity kinase in lines 93-94.

Comment 3:

On p. 15, lines 426-427, ‘tumor growth factor-b1’ should be ‘transforming growth factor-b1’.

Response 3: We corrected this error in the updated version.

Comment 4:

According to “Instructions for Authors”, the abstract should be a single paragraph.

Response 4: We corrected this error in the updated version.

Reviewer 2 Report

1. The manuscript is well written and organized based on the main hot topics in the area.

2. Please, be aware that Figure 13 is missing o misplaced. If not, then change (19, Figure 13) in line 439 and (20 Figure 13) in line 442 as (19, Figure 12) in line 439 and (20, Figure 12) in line 442, respectively .

Author Response

2. Please, be aware that Figure 13 is missing o misplaced. If not, then change (19, Figure 13) in line 439 and (20 Figure 13) in line 442 as (19, Figure 12) in line 439 and (20, Figure 12) in line 442, respectively .

Response 2: In fact, there was an error with transposing numbers. Figure 13 does not exist and we changed the references to "Figure 12". Please see the attachment.

Reviewer 3 Report

The review by Katzengruber and coll. reported the function of MKK4 protein kinase, its role on cancer development and liver regenerations and a classification of known inhibitors.

This review overall is clear, well written and very well focused on MKK4 inhibitors recently reported in the literature.

In my opinion only minor revisions are necessary, in detail:

ü   Please change references 1-4 with more recent references.

ü  Lines 50-51: please insert the name and corresponding clinical trials of MKK inhibitor.

ü  Please insert in paragraph 2 a Figure with a schematic representation of MKK4 protein.

ü  Please collect in one single figure figures 1-5.

ü  Please change the title of paragraph 3 in “MKK4 inhibitors”.

ü  For clarity in my opinion for one chemical class of inhibitors insert a specific paragraph (for example 3.1.1 9-H-pyrimido[4,5-b]ind-6-ol-scaffold, 3.1.2 7, ,3,4-trihydroxyisoflavone…3.1.3 genistein…..

ü  Regarding verumafenib, that showed off-target activity on MKK4 (line 361), please insert a IC50 values or percent of inhibition versus MKK4 and related reference.

ü  Please insert a list of abbreviations.

ü  Please follow MDPI author instruction for references format.

Author Response

  • Please change references 1-4 with more recent references.

Response 1: Two of those referenes are from 2021 and 2022 and we think they represent the general introducing information well.

  • Lines 50-51: please insert the name and corresponding clinical trials of MKK inhibitor.

Response 2: In the updated version, we inserted the corresponding clinical trial number EUDRA-CT No. 2021-000193-28, from the official press release by HepaRegenix (https://www.heparegenix.com/heparegenix-reports-positive-topline-results-of-its-phase-1-clinical-trial-of-hrx-0215-a-first-in-class-mkk4-inhibitor/). However, searching the  EU clinical trial register (https://www.clinicaltrialsregister.eu/ctr-search/search?query=2021-000129-28) did not match any clinical trial. According to the EU clinical trials register "Phase 1 trials conducted solely on adults and that are not part of an agreed paediatric investigation plan (PIP) are not publicly available" (https://eudract.ema.europa.eu/docs/guidance/EudraCT%20FAQ_for%20publication.pdf).

  • Please insert in paragraph 2 a Figure with a schematic representation of MKK4 protein.

Response 3: Even though there is a crystal structure of MKK4, binding AMP, we do not see any benefit since there is no structure of MKK4 co-crystallized with an inhibitor making any assumptions on a schematic model highly speculative.

Update Round2: We decided to implement a schematic representation of MKK4 (pdb: 3aln) monomer with respect to important structures for inhibitor design

  • Please collect in one single figure figures 1-5.

Response 4: In our opinion, the readability profits from the discussed structure visibly being directly on top of the text.

  • Please change the title of paragraph 3 in “MKK4 inhibitors”.

Response 5: In the updated version, we corrected this.

  • For clarity in my opinion for one chemical class of inhibitors insert a specific paragraph (for example 1.1 9-H-pyrimido[4,5-b]ind-6-ol-scaffold, 3.1.2 7, ,3,4-trihydroxyisoflavone…3.1.3 genistein…..

Response 6: In the updated version, we corrected this.

  • Regarding verumafenib, that showed off-target activity on MKK4 (line 361), please insert a IC50 values or percent of inhibition versus MKK4 and related reference.

Response 7: In the updated version, we added the percent of inhibition value from the literature and same reference in line 378.

  • Please insert a list of abbreviations.

Response 8: We agree, that some explanations for abbreviations were missing in the first version. In the updated version, we corrected this error and defined every non-classical abbreviation in the abstract, text, or figure. A list of abbreviations however is uncommon in this journal.

  • Please follow MDPI author instruction for references format.

Response 9: We altered the reference format according to the MDPI standards using their official reference style in the updated version.
